# Resting-State Neuronal Activity and Functional Connectivity Changes in the Visual Cortex after High Altitude Exposure: A Longitudinal Study

**DOI:** 10.3390/brainsci12060724

**Published:** 2022-05-31

**Authors:** Xinjuan Zhang, Taishan Kang, Yanqiu Liu, Fengjuan Yuan, Minglu Li, Jianzhong Lin, Jiaxing Zhang

**Affiliations:** 1Institute of Brain Diseases and Cognition, School of Medicine, Xiamen University, Xiamen 361102, China; zhang_xinjuan@foxmail.com (X.Z.); liuyanqiuchn@163.com (Y.L.); fegjuanyuan@163.com (F.Y.); 2Department of Radiology, Zhongshan Hospital of Xiamen University, School of Medicine, Xiamen University, Xiamen 361004, China; kkttss@126.com (T.K.); xmzshljz123@163.com (J.L.); 3School of Medicine, Xiamen University, Xiamen 361102, China; liminglu@stu.xmu.edu.cn

**Keywords:** high altitude, hypoxia, visual cortex, resting-state fMRI

## Abstract

Damage to the visual cortex structures after high altitude exposure has been well clarified. However, changes in the neuronal activity and functional connectivity (FC) of the visual cortex after hypoxia/reoxygenation remain unclear. Twenty-three sea-level college students, who took part in 30 days of teaching at high altitude (4300 m), underwent routine blood tests, visual behavior tests, and magnetic resonance imaging scans before they went to high altitude (Test 1), 7 days after they returned to sea level (Test 2), as well as 3 months (Test 3) after they returned to sea level. In this study, we investigated the hematological parameters, behavioral data, and spontaneous brain activity. There were significant differences among the tests in hematological parameters and spontaneous brain activity. The hematocrit, hemoglobin concentration, and red blood cell count were significantly increased in Test 2 as compared with Tests 1 and 3. As compared with Test 1, Test 3 increased amplitudes of low-frequency fluctuations (ALFF) in the right calcarine gyrus; Tests 2 and 3 increased ALFF in the right supplementary motor cortex, increased regional homogeneity (ReHo) in the left lingual gyrus, increased the voxel-mirrored homotopic connectivity (VMHC) value in the motor cortex, and decreased FC between the left lingual gyrus and left postcentral gyrus. The color accuracy in the visual task was positively correlated with ALFF and ReHo in Test 2. Hypoxia/reoxygenation increased functional connection between the neurons within the visual cortex and the motor cortex but decreased connection between the visual cortex and motor cortex.

## 1. Introduction

Nowadays, a large number of sea-level people migrate to high altitudes to travel or to work [1]. Altitude training has been employed for decades by elite endurance athletes in an attempt to enhance their sea-level performances [2], and exercise at altitude has been gaining popularity in recent decades [3]. Since hypoxia is the greatest challenge of living in high-altitude areas, people living in high-altitude areas inevitably suffer from hypoxia stress. When people who have migrated to high altitudes return to sea level, their bodies experience a process from hypoxia to reoxygenation. The brain is one of the organs in the body with the highest oxygen consumption, and thus, is particularly vulnerable to hypoxia. Moreover, the brain is the structural basis of high-level neural activities and the high-level center for regulating body activities, and thus, clarification of the structure and physiological function of the brain at high altitude is helpful to understanding the behavior of human high-altitude activities.

The information processed by the visual cortex accounts for more than 70% of all brain sensory information. Firstly, the visual cortex dynamically processes received sensory information on a millisecond time scale [4], and its cells are the most active and consume the most energy. Therefore, it is very vulnerable to hypoxia. Secondly, previous studies have shown that high-altitude environments have damaging consequences to vision, contrast sensitivity, visual field, and color vision, suggesting that the retinal photosensitive function and visual information transmission function are reduced [5,6,7,8]. A decrease in visual information input inevitably leads to adaptive adjustment of the visual cortex. Thirdly, the visual cortex receives fiber projections from the contralateral retina through the lateral geniculate body and also has internal connections and feedback connections in the primary visual cortex, and thus, it shows more precise organization rules than other brain regions [9]. The fine arrangement of the visual cortex makes its recovery ability limited after injury, but the compensation mechanism of the visual cortex is very effective [10]. Finally, the visual cortex of adults retains a “steady-state plasticity” [11]. To sum up, hypoxia has effects on different functional brain regions, and research on the visual cortex can best reflect the effect and remodeling of hypoxia on the brain.

In recent years, the structural and functional changes in the brains of individuals at high altitudes have been examined using magnetic resonance imaging (MRI) techniques. Although multiple regions of the brain are affected during high altitude exposure, the more consistent findings have involved the occipital visual cortex, covering the lingual gyrus, fusiform gyrus, calcarine, and precuneus. In the visual cortex of high altitude residents, decreased gray matter volume [12] and decreased cerebrovascular reactivity [13] have been found in the descendants of the Han population who have immigrated to high altitude for several generations; decreased cerebral blood flow [14] and cortical thickness [15] have been found in high-altitude Tibetan natives. A study on soldiers who had been garrisoning the border for two years in the Himalayas showed that the functional connection of the bilateral visual cortex [16] and the amplitude of low-frequency fluctuation (ALFF) of the visual cortex were significantly enhanced in resting state within one week after they returned to the plain [17]. A dynamic tracking electroencephalography (EEG) study on soldiers showed that after 30 days of high altitude exposure, alpha and beta power increased in the occipital cortex; after soldiers returned to sea level within 1 week, beta power continued to increase [18]. Another study showed that the occipital visual cortex thickened after 1 month of exposure to high altitude and recovered to the baseline level after participants returned to sea level for 2 months, but there were still abnormalities in the white matter [19]. A study found that cerebral alterations were reversible after an exposure of 7 day at a high altitude of 4554 m and then returning to sea level for 3.5 months [20]. These studies mainly focused on chronic high altitude exposure and relatively less work has been done based on acute high altitude exposure and to clarify whether these changes were reversible or not after returning to sea level through longitudinal studies. Acute high altitude exposure can better reflect the effect of brain hypoxia and is less disturbed by additional variables. Moreover, it is unknown if changes in the visual cortex function after high altitude exposure lead to changes in functional connectivity with other brain regions.

Resting-state functional magnetic resonance imaging has the advantage of directly observing the spontaneous neurological activity in the brain (Biswal et al., 2010), which is widely used for studying brain functions. In resting-state fMRI, ALFF has been shown to be a valuable parameter that reflects the intensity of spontaneous nerve activity [21], regional homogeneity (ReHo) has been employed to analyze the similarities or coherence of intraregional spontaneous BOLD signal fluctuations [22], voxel-mirrored homotopic functional connectivity (VMHC) has been employed to investigate inter-hemisphere homotopic connections [23], and FC has been employed to describe the correlation of brain spontaneous functional activities in the same time series [24]. Thus, this study aimed to analyze the neuronal activity and FC in the visual cortex after hypoxia exposure. Based on previous studies, we hypothesized that hypoxia/reoxygenation mainly changes the neuronal activity in the visual cortex, and the functional connections between the visual cortex and other brain regions are also affected.

## 2. Material and Methods

### 2.1. Participants

Twenty-three participants were recruited from the Volunteer Teaching Team of Xiamen University (11 female/12 male, mean age 19.43 ± 0.88). Every year, different students go to Tibet to teach during the summer vacation in this Volunteer Teaching Team. The participants were all born and lived at sea level (<500 m) without any prior exposure to high altitude. They took part in 30 days of teaching on the Qinghai-Tibetan plateau (4300 m). We chose a medium altitude of 4300 m [25], which was more suitable for immigration. All participants had a normal or corrected-to-normal vision and had successfully finished teaching work, without the use of supplementary oxygen. None of them had a history of psychiatric disorders. A written informed consent from each participant was obtained before the experiment. The study was conducted in accordance with the Declaration of Helsinki, and the experimental protocol was approved by the Ethical Committee of Xiamen University (XDYX2016013). Twenty sea-level controls were also scanned and rescanned at an interval of 30 days, and no observed differences in any cerebral measurements were found between the two MRI scans.

The participants took 3 days to travel from sea level (63.2 m) to Lhasa (3650 m, Tibet, China) by train. After staying at Lhasa for 4 days, they completed a 4-h bus drive to Dangxiong city (4300 m, Tibet, China). After 30 days of teaching, they returned to Lhasa for 4 days, and then descended to sea level. The participants did not smoke and were not allowed access to alcohol.

Routine blood tests, behavior tests, and MRI scanning were performed before the participants ascended to high altitude (Test 1) and after they returned to sea level for 7 days (Test 2), and after they had returned to sea level for 3 months (Test 3). Previous research has shown that we basically acclimatize to high altitude after one month, and returning from a high altitude to sea level requires about 20 days for adaptation to occur [26]. Therefore, we chose Test 1 as the baseline, Test 2 as a time point when the volunteers had not fully adapted to sea level after high altitude exposure, and Test 3 as a time point when the volunteers had fully adapted to sea level after high altitude exposure. Each test was completed within 5 days.

### 2.2. Routine Blood and Behavior Tests

Blood samples in three tests were taken in the morning between 7:00 and 7:30 at the the Zhongshan Hospital of Xiamen University.

In this study, we focused on the visual cortex, therefore, we added the visual behavior test, which adopted a clock paradigm including visual space and color tasks, similar to previous research [27]. Participants were asked to judge the angle and color of the minute hand and second hand on a clock according to the prompts, while the reaction time and accuracy were recorded. All the stimuli were presented and controlled using the E-prime software system (Version 3.0, Psychology Software Tools, Inc., Pittsburgh, PA, USA).

### 2.3. MRI Data Acquisition

Brain images were acquired on a Tim Trio 3T scanner (Siemens, Erlangen, Germany) with an 8-channel phase array head coil at the MRI Center in the Zhongshan Hospital of Xiamen University. We acquired T1 and T2 images, and two experienced radiologists viewed the images to exclude participants with space-occupying lesions and cerebrovascular diseases. A 3D structural MRI was acquired using a T1-weighted MPRAGE sequence (TR/TE = 1900/2.7 ms, FOV = 25 × 25 cm, average = 1, matrix = 256 × 246, flip angle = 9°, slice thickness = 1.0 mm, voxel size = 1 × 1 × 1 mm). Resting-state fMRI images were obtained using an echo-planar imaging sequence with the following parameters: TR/TE = 3000 ms/30 ms, flip angle = 90°, matrix = 64 × 64, voxel size = 3.4 × 3.4 × 3.75 mm, FOV = 24 × 24 cm, slices = 39, slice thickness = 3 mm, and 6 min.

### 2.4. Resting-State fMRI Data Preprocessing

The preprocessing of resting-state fMRI data was conducted in MATLAB 16.0 (https://www.mathworks.com, accessed on 19 January 2021) by using the statistical parametric mapping method (SPM 12, https://www.fil.ion.ucl.ac.uk/spm/software/spm12/, accessed on 19 January 2021) and the Data Processing Assistant for Resting-State fMRI (DPARSF, V5.1, http://rfmri.org/DPARSF, accessed on 19 January 2021) [28] module of a toolbox for Data Processing Analysis of Brain Imaging (DPABI, http://rfmri.org/dpabi, accessed on 19 January 2021) [29]. The initial 10 time points were removed to eliminate the magnetic saturation effect and adapt to the scanning environment of participants. Then, slice timing and head motion corrections were performed (the head motion was <2.5 mm in any x, y, and z direction and <2.5° in any angular dimension). After data quality thresholds were set, 3 participants were excluded from further analysis. After head motion correction, T1 images were co-registered to the mean functional image for each participant. The co-registered functional images were normalized to the Montreal Neurological Institute (MNI) space and resampled to 3 × 3 × 3 mm Finally, spatial smoothing was performed with a Gaussian kernel of 4 mm full-width half-maximum Gaussian kernel (FWHM), and linear trends within the time series were removed.

### 2.5. Analyses of Resting-State fMRI

The ALFF, ReHo, and VMHC were calculated using the DPABI. Each voxel’s time series is transformed into frequency domain by Fourier transform, and then the power spectrum is obtained. Subsequently, the square root of each frequency power spectrum is calculated according to the frequency band (usually 0.01–0.1 Hz), and the mean value is ALFF. We performed temporal bandpass filtering (0.01–0.1 Hz) on the all-time series. The ReHo value was divided by the global mean Kendall coefficient of concordance value for standardization. Finally, spatial smoothing was applied for the ReHo analysis with a 4 mm × 4 mm × 4 mm full-width half-maximum isotropic Gaussian kernel. The VMHC values were Fisher Z transformed and smoothed (4 mm FWHM). FC was examined with a seed-voxel correlation approach in DPARSF. The peak of the differential brain region activated by ReHo was used as the seed point (coordinates: x = −24, y = −93, and z = 6) in the left lingual gyrus with a radius of 6 mm. A voxel-wise correlation was then computed between the time series of the seed reference and those of all brain voxels outside of the seed region. A Fisher r-to-Z transformation was used to transform the correlation coefficients to Z values.

### 2.6. Statistical Analysis

The hematological parameters and behavior results were analyzed using repeated-measures analysis of variance (ANOVA). Whole-brain voxel-wise-based repeated-measures ANOVA was employed, with the head motion parameters as covariates. Statistical analyses of the fMRI data were performed using the statistical module of DPABI. Repeated-measures ANOVA and multiple comparisons were applied to compare the differences in the ALFF, ReHo, VMHC, and FC among Tests 1, 2, and 3. We used the 5000 permutations test with threshold-free cluster enhancement (TFCE) as a strict multiple comparison correction [30]. Significant clusters were detected when *p*  <  0.05 with family-wise error (TFCE-FWE) correction and a cluster size ≥60 voxels. The mean image values of each individual were extracted, and the group difference was analyzed using one-way repeated measure ANOVA with post hoc in Bonferroni correction.

## 3. Results

### 3.1. Routine Blood Results

The blood routine analysis results in the three time-point tests are summarized in Table 1. There were significant differences in hematocrit, hemoglobin, and red blood cell count. The post hoc test showed that the hematocrit, hemoglobin, and red blood cell count were significantly increased in Test 2 as compared with Tests 1 and 3. The routine blood results in Test 3 returned to the baseline level.

### 3.2. Behavior Results

The mean reaction times and mean accuracy in performing the clock task in the three time-point tests are summarized in Table 2. There were no significant differences in mean reaction times and mean accuracy among the three time-point tests.

### 3.3. ALFF Values

The ALFF changes among three time-point tests are shown in Figure 1 and Table 3. There were significant differences of ALFF values among the tests in the right calcarine (*F* = 20.03, *p* < 0.05, FWE correction), right precentral gyrus (*F* = 11.24, *p* < 0.05, FWE correction), and right supplementary motor cortex (*F* = 12.44, *p* < 0.05, FWE correction) (Figure 1A). The post hoc test showed that there was a higher ALFF value in the right calcarine gyrus in Test 3 as compared with Test 1, a higher ALFF value in the right precentral gyrus in Test 2 as compared with Test 1, and a higher ALFF value in the right supplementary motor cortex in Tests 2 and 3 as compared with Test 1 (Figure 1B).

### 3.4. ReHo Values

The ReHo changes among three time-point tests are shown in Figure 2 and Table 4. There were significant differences among the tests in the left lingual (*F* = 19.93, *p* < 0.05, FWE correction) (Figure 2A). The post hoc test revealed a higher ReHo value in the left lingual gyrus in Tests 2 and 3 as compared with Test 1 (Figure 2B).

### 3.5. VMHC Values

The VMHC changes among three time-point tests are shown in Figure 3 and Table 5. There were significant differences among the tests in the motor cortex (*F* = 16.73, *p* < 0.05, FWE correction) (Figure 3A). The post hoc test indicated a higher VMHC value in the precentral gyrus extending to the postcentral gyrus in Tests 2 and 3 as compared with Test 1 (Figure 3B).

### 3.6. FC Values

The FC changes among three time-point tests are shown in Figure 4 and Table 6. There were significant differences among the tests in the left postcentral gyrus (*F* = 14.23, *p* < 0.05, FWE correction) (Figure 4A). The post hoc test demonstrated lower FC values between the left lingual gyrus and left postcentral gyrus in Tests 2 and 3 as compared with Test 1 (Figure 4B).

### 3.7. Correlations

The correlations of behavior tests with ALFF values in the right calcarine and ReHo values in the left lingual gyrus are shown in Figure 5. The mean accuracy in performing the clock task was positively correlated with ALFF values in right calcarine (*r* = 0.583, *p* = 0.018) and ReHo values in the left lingual gyrus (*r* = 0.589, *p* = 0.016) in Test 2.

## 4. Discussion

In this study, the hematological parameters, visual behavior test, and spontaneous brain activity of the visual cortex after hypoxia/reoxygenation were investigated. The hematocrit, hemoglobin concentration, and red blood cell count increased after high altitude exposure. We found neuronal activity and internal connection were increased in the visual cortex and motor cortex, but decreased the connection between the visual cortex and motor cortex (Figure 6). Additionally, the visual behavior results were positively correlated with neuronal activity after reoxygenation for 7 days. In addition, after 1 month of hypoxia and reoxygenation for 3 months, the routine blood test results returned to the baseline level, but the neuronal activity and FC of the visual cortex did not return to the baseline level.

At high altitudes, a drop in air pressure leads to a decrease in the partial pressure of oxygen and a decrease in the oxygen saturation of hemoglobin [31]. To increase tissue oxygen supply, hypoxia stimulates erythropoiesis by complex molecular mechanisms to elevate the concentration of hemoglobin [32]. Previous studies have shown that hematocrit level and hemoglobin concentration increased and were stable parameters for acclimatization at high altitude [33]. Our routine blood test demonstrated increased hematocrit level, hemoglobin concentration, and red blood cell count after reoxygenation for 7 days, but they could return to the baseline level after reoxygenation for 3 months. BOLD represents a combination of vascular and neural activities. There were no hematological changes after 1 month of high altitude exposure and returning to sea level for 3 months, but changes in BOLD were still observed, indicating that the effect of hypoxia on nerve activity was longer than that on blood. The reason why hypoxia has a longer lasting effect on brain activity as compared with the blood may be due to the fact that the central nervous system is more sensitive to hypoxia, with the highest oxygen consumption. However, increased vasculature after 2–8 weeks of hypoxia [34,35] has been shown to maintain an optimal continuous supply of oxygen and nutrients to the brain to account for the hypoxia environment [36]. In addition, a positive correlation between vascular density and ALFF signal has been reported [37]. Therefore, we speculated that the increase in ALFF may be related to an increase in vasculature.

Hypoxia/reoxygenation increased ALFF in the occipital visual cortex, precentral gyrus, and supplementary motor cortex, which may be associated with an increase in vasculature. Moreover, previous studies have found that high altitude exposure increased the cortical thickness in the visual cortex and motor cortex which may also have been due to increased vascular density [19,38]. Studies in animals have been consistent with our results, showing that the increase in blood vessels caused by hypoxia did not return to the baseline level after reoxygenation for a while. Previous studies have found that cerebral microvessels were significantly increased after 1 and 3 weeks of hypoxia, and remained elevated after 3 weeks of normoxia in rats [35]; after 8 weeks of hypoxia and reoxygenation for 8 weeks, the blood vessel area was still increased in rats; cyclooxygenase-2 and angiopoietin-2 were both elevated during hypoxia for 21 days as well as subsequent re-oxygenation for 21 days in mice [36]. In a natural environment, the visual cortex continuously receives external information which is the most active, and the motor cortex continuously outputs motion signals. The visual cortex and the motor cortex may require greater oxygen consumption than other cortices, and therefore, are more sensitive to hypoxia. Increased neuronal activity after high altitude exposure is consistent with previous studies. Zhao et al. recorded EEG activity in resting state at four time points: before ascending to high altitude, during 7 days and 30 days at 3800 m altitude, and 7 days after return to sea level. They found a persistent increased beta power in the parietal cortex and occipital cortex during the first 7 days at high altitude and after 7 days return to sea level [18]. Beta activity is indicative of background excitation that involves a frequency potentiating mechanism at the synaptic level in a network [39]. Another study led by Zhang et al. found that the participants who had garrisoned the frontiers at high altitude for 2 years and then returned to sea level for 7 days had increased ALFF within the bilateral occipital cortices [17]. Increased ALFF in the visual cortex was found in both short-term and long-term hypoxia, suggesting that the visual cortex plays an important role in both long-term and short-term adaptation to hypoxia. After 3 months of reoxygenation, the nerve cell activities of the visual cortex and motor cortex did not return to the baseline level, indicating that the proliferation of blood vessels caused by hypoxia could not be eliminated after 3 months of reoxygenation. Previous studies have found that 7 days of high altitude exposure can recover in 3.5 months after returning to the plains [20], therefore, we speculate that after 1 month of high altitude exposure, it would take more than 3 months to recover.

Hypoxia/reoxygenation only increased ReHo in the occipital visual cortex, indicating increased neural coherence, and may reflect a compensatory role in vision deficits. In support of our findings, some previous studies have also shown decreased visual function in hypoxia exposure [6,8]. Similarly, event-related potential studies have found that chronic high altitude exposure decreased event-related desynchronization in the parietal and occipital regions [40] and decreased P50 delay activity amplitudes that reflected a predominantly pre-attentional inhibitory filter mechanism in regions [41] included by the visual mental rotation task. Considering our results, the increase in ReHo may be a compensatory mechanism for the visual cortex that needs to call for more cognitive resources to process information. These findings suggest that the visual cortex may be more sensitive to hypoxia and plays an important role in hypoxia that is worthy of further investigation. In contrast, a previous neuroimaging study showed that in college students who had high altitude exposure for 2 years, ReHo increased in the bilateral hippocampi and decreased in the putamen, bilateral superior temporal gyri, bilateral superior parietal lobules, anterior cingulate gyrus, and medial frontal gyrus [42]. Another study showed that in the participants who had garrisoned the frontiers at high altitude for 2 years, ReHo increased in the right inferolateral sensorimotor cortex [43]. Different from these two studies, we only found increased ReHo in the occipital visual cortex. We inferred that the differences in results may be attributed to the different high altitude exposure times; our study design was acute high altitude exposure for 1 month, which was different from chronic high altitude exposure for 2 years in these two studies. Moreover, the accuracy of the visual clock task under color resolution conditions was positively correlated with ALFF and ReHo values in the occipital visual cortex, showing that resting-state neuronal activity may be responsible for the behavior changes.

Interestingly, the ALFF and ReHo results are presented with lateralized differences in the visual cortex, furthermore, the lateralization positions are different. To the best of our knowledge, hemispheric specialization is an important principle of cortical processing for vision [44]. ALFF enhancement on the right side of the visual cortex indicates that the oxygen supply is stronger than that on the left side, and therefore, the left side of the visual cortex needs to call for more resources for compensation during processing, thus, ReHo enhancement is on the left side of the visual cortex. However, no significant differences were found among the three visual behavior tests, which may be caused by the practice effect of the task. Practice effect refers to the phenomenon that with an increase in the number of experiments, a subject’s proficiency in the operation of experimental tasks is gradually improved, and the accuracy of response gets better and better. Practice strengthens new schemas such that stimuli become capable of automatically engaging associated processes and responses [45]. However, after high altitude exposure, accuracy has a decreasing trend and the reaction time has an increasing trend, which is consistent with the visual impairment caused by high altitude exposure [6].

In our present study, the increased VMHC values in the precentral gyrus after hypoxia/reoxygenation may be the mechanism underlying the improvement in altitude training performance. Increased VMHC in the precentral gyrus was also found in professional rowers who received transcranial direct current stimulation, and the VMHC values were positively correlated with the measures of athletic performance [46]. A previous study also found that increased connectivity within the sensorimotor cortices was potentially involved with the translation of thought planning into motor programs [47]. Decreased FC between the occipital visual cortex and postcentral gyrus after reoxygenation might indicate a decrease in visual-motor function. Coordination of the visual cortex and the motor cortex is involved in many tasks that apply visual-motor function. Visual-motor function is critical in the integration of visual perception and motor skills [48]. The visual cortex and motor cortex show co-activity during a visual tracking task [49]. The spontaneous activity in the visual cortex is associated with the motor cortex in the resting state [50]. Evidence has shown functional connectivity in several diseases such as early blindness [51] and migraine [52]. Our findings indicated that hypoxia/reoxygenation might present with abnormal visual-motor function, which suggests that altitude training may not be suitable for sports requiring hand-eye coordination, such as table tennis, badminton, and so on. Overall, we considered that the FC of the visual cortex and motor cortex were related to the physiological change of high altitude exposure; future studies are needed to elucidate this speculation.

Some limitations of the study must be recognized. First, we did not further confirm the neuroimaging changes by excluding the effects of diet and culture. Second, we did not follow a longer period to clarify the time point of recovery after high altitude exposure for 1 month.

## 5. Conclusions

This is a systematic longitudinal study after hypoxia/reoxygenation. We found that hypoxia/reoxygenation increased resting-state neuronal activity in the visual cortex and motor cortex, and reduced the connection between the visual cortex and motor cortex. Increasing internal connection in the motor cortex may be the mechanism for the improvement of altitude training performance. After 1 month of hypoxia and reoxygenation for 3 months, the neuronal activity did not return to the baseline level. The observation time needs to be further extended and the internal mechanism needs to be verified by animal models. Future research needs to further study the mechanism at the tissue, cell, and molecular levels through animal models. The study of hypoxia/reoxygenation is conducive to the formulation of policies related to plateau activities.

## Figures and Tables

**Figure 1 brainsci-12-00724-f001:**
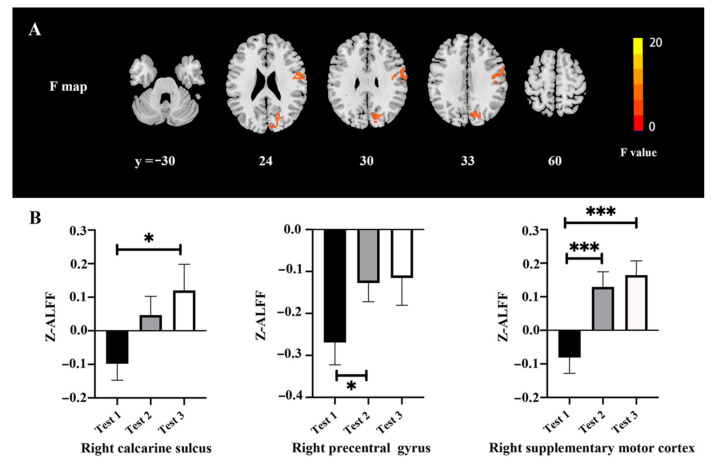
ALFF changes among the three time points: (**A**) Difference of ALFF maps among the three time points; (**B**) extracted clusters of significant ALFF alteration in three time points. * *p* < 0.05 and *** *p* < 0.001.

**Figure 2 brainsci-12-00724-f002:**
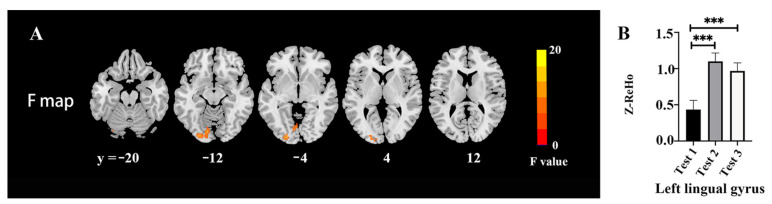
ReHo changes among the three time points: (**A**) Difference of ReHo maps among the three time points; (**B**) extracted clusters of significant ReHo alteration in three time points. *** *p* < 0.001.

**Figure 3 brainsci-12-00724-f003:**
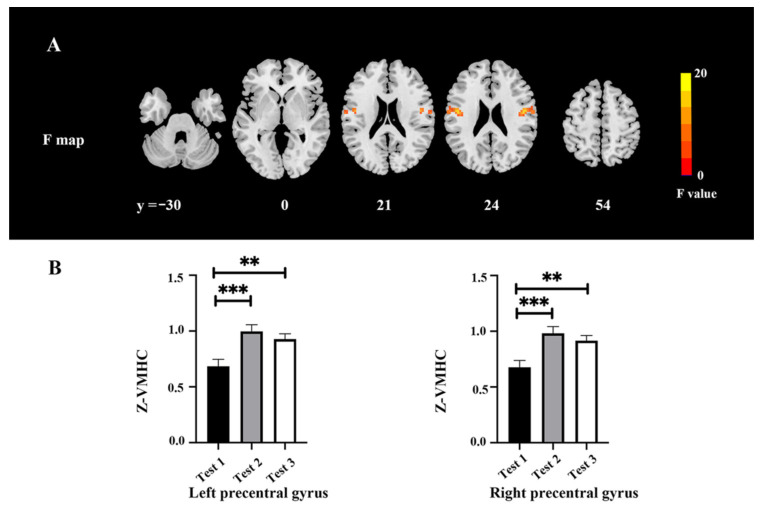
VMHC changes among the three time points. (**A**) Difference of VMHC maps among the three time points. (**B**) Extracted clusters of significant VMHC alteration in three time points. ** *p* < 0.01 and *** *p* < 0.001.

**Figure 4 brainsci-12-00724-f004:**
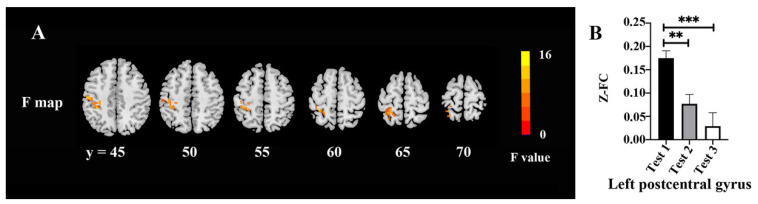
FC changes among the three time points: (**A**) Difference of FC maps among the three time points; (**B**) extracted clusters of significant FC alteration in three time points. ** *p* < 0.01 and *** *p* < 0.001.

**Figure 5 brainsci-12-00724-f005:**
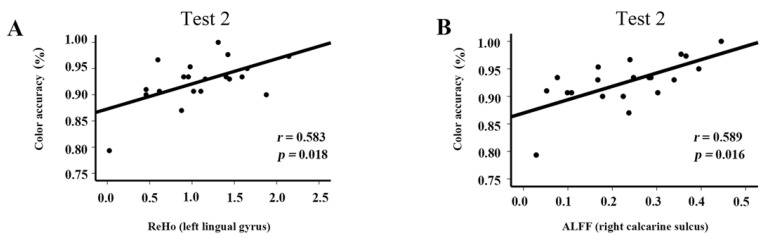
Correlations between spontaneous brain activity and behavior results in Test 2: (**A**) Correlation between color accuracy in the visual task and ReHo in the left lingual gyrus; (**B**) correlation between color accuracy in the visual task and ALFF in the right calcarine gyrus.

**Figure 6 brainsci-12-00724-f006:**
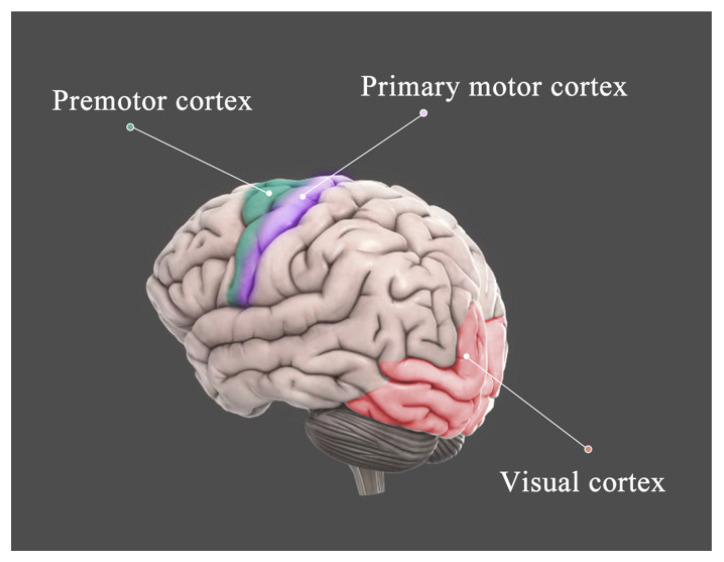
The schematic figure of the visual cortex and the motor cortex.

**Table 1 brainsci-12-00724-t001:** Hematological results.

	Test 1	Test 2	Test 3	*F*	*p*
Hematocrit (%)	40.69 ± 3.17	42.87 ± 4.62	39.59 ± 3.26	13.77	<0.0001
Hemoglobin (g/L)	133.10 ± 14.03	141.95 ± 18.22	132.20 ± 13.78	14.92	<0.0001
Red blood cells count (10^12^/L)	4.77 ± 0.50	5.02 ± 0.75	4.65 ± 0.70	16.50	<0.0001

**Table 2 brainsci-12-00724-t002:** Behavior results in Tests 1/2/3.

	Angle ACC (%)	Color ACC (%)	Angle RT (ms)	Color RT (ms)
Test 1	90.86 ± 5.25	94.71 ± 3.14	575.35 ± 96.63	527.04 ± 81.88
Test 2	86.77 ± 4.36	92.59 ± 4.46	618.52 ± 85.82	549.92 ± 92.06
Test 3	89.66 ± 6.50	92.99 ± 3.58	562.87 ± 84.23	522.16 ± 98.89

ACC, accuracy; RT, reaction time.

**Table 3 brainsci-12-00724-t003:** Regional information of changed ALFF in repeat ANOVA.

Area	Broadmann Area	Volume (Voxels)	Talairach			*F* (Peak)
			X	Y	Z	
Occipital/Temporal/Calcarine_R/Lingual_R/Precuneus	17/18/19/37	920	48	−57	6	20.03
Precentral_R	6/4	260	60	3	36	11.24
Supp_Motor_Area_R	6	168	6	3	78	12.44

R, right.

**Table 4 brainsci-12-00724-t004:** Regional information of changed ReHo in repeat ANOVA.

Area	Broadmann Area	Volume (Voxels)	Talairach			*F* (Peak)
			X	Y	Z	
Occipital/Lingual_L/Fusiform_L/Calcarine_L	17/18/19	179	−24	−93	6	19.93

L: left.

**Table 5 brainsci-12-00724-t005:** Regional information of changed VMHC in repeat ANOVA.

Area	Broadmann Area	Volume (Voxels)	Talairach			*F* (Peak)
			X	Y	Z	
Precentral_L/Postcentral_L	6/4	93	−48	−46	24	16.73
Precentral_R/Postcentral_R	6/4	104	48	−46	24	16.73

R, right; L, left.

**Table 6 brainsci-12-00724-t006:** Regional information of changed FC in repeat ANOVA.

Area	Broadmann Area	Volume (Voxels)	Talairach			*F* (Peak)
			X	Y	Z	
Postcentral_L	6	174	−51	−18	45	14.23

L, left.

## Data Availability

The data presented in this study are available on request from the corresponding author. The data are not publicly available due to the data contains the subject’s personal privacy.

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
