# Peer review of "Resting-State Neuronal Activity and Functional Connectivity Changes in the Visual Cortex after High Altitude Exposure: A Longitudinal Study"

_brainsci, 2022, doi:10.3390/brainsci12060724_

Round 1
Reviewer 1 Report
Minor
- Introduction
Previous studies are mentioned here, but there is only one reference, other sources should be added. (Secondly, previous studies have shown that the high-altitude environment has damage consequences to vision, contrast sensitivity, visual field, and color vision, suggesting that the retinal photosensitive function and visual information transmission function are reduced [5]).
- 1. Participants:
- References should be added on determining the altitude to be ascended, determining the duration of stay at altitude, and how long after the tests should be done after descending to sea level.
- The reasons for performing test-1, test-2 and test-3 on the days on which they were determined should be explained and/or should be based on the literature.
Major
- In the results, blood values are presented according to gender difference. However, in the presentation of the results of the other parameters examined by the research, the gender difference was not emphasized. What is the purpose of including gender difference in blood values? The results of blood values given with gender difference are presented in the discussion section without gender difference. Can researchers present other results by considering the gender difference?
- (2. Routine blood and behavior tests): The researchers applied a clock task to the participants at 3 different times. In the Results section, they presented the Angel ACC/RT and color ACC/RT results obtained from the clock task. Although the researchers presented these results, they did not sufficiently emphasize the reasons for applying this test in the article. Findings from the clock task were not discussed in the discussion part of the article. It is necessary to discuss the results from the clock task in the Discussion section.
Reviewer 2 Report
I think this research is meaningful in that it tried to clarify how the neuronal activity and functional connectivity (FC) of the visual cortex after hypoxia-reoxygenation, which had not been clarified so far. However, wasn't there any ethical problem in selecting volunteers from the campus and conducting experiments in situations that could cause nerve damage? I wonder that the content of informed consent and the review by the Institutional Review Board.
Reviewer 3 Report
Within this manuscript, the author investigated the effect of altitude on different parameters of the brain. The paper is overall technically sound, interesting, and easy to read. The only thing that should be corrected is the way of reporting the p-value in the tables, for a small p-value it is better to write <0.001, or where there is a need to emphasize the small amount of p-value, <0.0001 can be used.
Author Response
Point 1: Within this manuscript, the author investigated the effect of altitude on different parameters of the brain. The paper is overall technically sound, interesting, and easy to read. The only thing that should be corrected is the way of reporting the p-value in the tables, for a small p-value it is better to write <0.001, or where there is a need to emphasize the small amount of p-value, <0.0001 can be used.
Response 1: Thanks for your suggestion. We have revised the way of reporting the p-value in this manuscript.
Reviewer 4 Report
The presented study reports alterations in spontaneous brain activity in the visual and motor cortex of subjects when exposed to high altitude for 30 days and later returned to sea level. The study used various fMRI metrics to test resting-state brain activity and connectivity at 3 time points. Blood tests were also conducted to determine the RBC count and hemoglobin levels.
The study overall is clear and scientifically sound. However, a number of concerns exist, that if satisfactorily addressed would improve the presentation of the findings and gain access to a wider (non-specialized audience).
Comments:
1) Study cohort: Is the study cohort from which data for this study was obtained is the same one described in a previous study by the same authors (DOI: 10.14814/phy2.15036)?
If yes, the itinerary of the plateau trip is slightly different. In this study subjects have a 4 days layover in Lhasa, whereas in the previous study the layover was only for 1 day. Can the authors clarify and insert their comments in the main text.
2) It would be very beneficial for the reader if the authors can generate a schematic figure that shows the location of all the tested brain sub-regions in relation to the visual and motor cortex.
4) it will help the reader greatly particularly those whom are interested but outside the field of neuroimaging to include an introductory sentence explaining which aspect of brain physiology is being tested with each imaging metric (ALFF, VHMC, etc..)
5) The authors report no significant alteration in behavioral tests in this study, however, the same tests when conducted under similar conditions revealed significant decline in the spatial and non-spatial visual tasks (DOI: 10.14814/phy2.15036), can the authors comment on this discrepancy
6) Can the authors speculate why hypoxia has a longer lasting effect on brain activity compared to the blood
7) “In support of our findings, several previous studies also showed decreased visual …..[31]”
the cited study [31] did not test vision impairment or visual cortex activity, but rather visual memory which is a different function, the authors can revise the citation.
8) “There is no difference in our visual behavior test, which may be due to the practice effect.” Can the authors elaborate more on the "practice effect", is this a known phenomenon? and include supporting references.
9) “…professional rowing athletes who accepted through transcranial direct current stimulation,…” this sentence requires revision
10) Figures and tables:
a) panel A in all figures:
- What does F value indicate?
- What does the number below each axial brain image signify?
- are these images obtained from one subject or several subjects?
- have they been obtained at the same or different time points
b) Table 1: the units of the measured blood components are missing, also “Red Blood Cell” should read Red blood cells count
- what does the italicized “F” signifies?
c) Table 3: what does BA signify?
d) Tables 3-5:
- Is there a reason for not listing the volume and coordinates of each region individually?
- Why some regions are in bold?
e) Figure and Table 4: the values for lingual gyrus are not presented
Round 2
Reviewer 2 Report
I have confirmed the revised manuscript.
I think the manuscript has been sufficiently improved to warrant publication in Brain Sciences.